# An AI-First Proof of Concept: Simulating and Refining a Teach-Back Protocol for Dialogic Learning in Programming Education

**GPT-4o; Marta Valentini**
Department of Information Engineering and Computer Science (DISI)
University of Trento
Trento, Italy
`marta.valentini-1@unitn.it`

## Abstract

This AI-first proof-of-concept investigates whether a generative agent can design and enact a dialogic Teach-Back protocol for novice programmers. We ran three controlled simulations on a C++ for-loop task and conducted a lightweight thematic analysis of the resulting dialogues. The analysis surfaced recurrent misconceptions and metacognitive moves and informed three refinements to the protocol: a dedicated Slow-Thinking phase, incremental probing to manage cognitive load, and a Transfer phase to prompt forward-looking application and self-explanation. Findings are illustrative from AI self-play and bounded to this simulated setup; they indicate how an AI can act as a dialogic partner—rather than an answer engine—supporting conceptual clarification and metacognitive reflection. We release prompts and anonymized logs to enable inspection and reuse. A classroom deployment in Programming 1 is planned, positioning the protocol as a transparent, method-driven template for integrating AI into learning environments.

## 1 Introduction

Generative AI, when unstructured, often replaces cognitive effort instead of fostering it. Recent work has shown that unguided use of LLMs in education can lead to surface-level learning and skill decay [1, 2]. In programming education, this may lead to surface-level learning, especially among novice learners. The increasing integration of large language models (LLMs) in university settings presents both opportunities and risks: while they can support student reasoning, they can also encourage dependence unless guided by structured pedagogical scaffolds [1, 2].

This study explores whether and how AI agents can scaffold conceptual understanding and metacognitive awareness through the Teach-Back method - a strategy in which students are asked to explain concepts as if teaching them to others, reinforcing internalization and clarity of understanding [3, 4, 5].

In particular:

- RQ1: Compared to an unguided AI baseline, how does the *Teach-Back* protocol modify the learning-oriented qualities of the interaction (structure, student role, error handling, metacognition)?
- RQ2: Which patterns of conceptual misunderstanding and metacognitive reflection emerge within these protocol-guided interactions?
- RQ3: How do the insights derived from the simulations inform refinements to the protocol?

The agent planned and conducted the entire study pipeline, from protocol design to simulation and analysis, under human pedagogical supervision.

This paper is a design-led proof-of-concept based on three controlled simulations. Our aim is to make the interaction design transparent and to distill design implications, not to estimate causal effects or claim generalizability to classroom outcomes. Analyses are qualitative and illustrative; we release prompts and anonymized logs for inspection and reuse. This is not a controlled efficacy study, a statistical comparison, or a definitive evaluation of learning gains, but a bounded empirical case and a method-driven template to be evaluated in future deployments.

## 2  Related Work

The present study is situated at the intersection of dialogic pedagogy, generative AI, and programming education. Previous research has demonstrated both the risks and opportunities of using AI in higher education. Unstructured use of LLMs can hinder deep learning and accelerate skill decay [6, 1, 2]. However, when embedded within structured pedagogical strategies, AI tools can enhance reflective thinking and metacognition [7, 8, 9, 10, 4].

The Teach-Back method, although well-established in health education and other STEM disciplines [3, 4, 5], remains underexplored in computer science education, making this application a novel contribution to AI-mediated pedagogical design. Kovács et al. [3] and Manzano-Vázquez [5, 11] applied teach-back principles in AI and general higher education contexts, highlighting its potential for engagement and retention. Sherman et al. [4] further demonstrate its use in bridging learning gaps.

Recent advances in AI-enhanced critical thinking and dialogic interaction frameworks support the design of conversational agents that stimulate reasoning rather than provide answers. Work by Favero et al. [12], Fakour and Imani [13], and Tankelevitch et al. [9] demonstrate how Socratic or epistemically challenging AI agents can enhance metacognitive effort and learning outcomes.

This study builds on these foundations by combining dialogic AI scaffolding with a revised Teach-Back protocol in a computer science education setting, contributing both a refined pedagogical model and a proof-of-concept implementation.

## 3  The Teach-Back Protocol (Initial Five-Phase Version)

At the center of this study is the design of the *Teach-Back* protocol, a structured dialogic sequence intended to scaffold student reasoning rather than provide direct answers. The protocol was adapted from established formative-feedback frameworks and refined through iterative simulations. The design is explicitly dialogic: the AI does not explain in the student's place but prompts, listens, and follows up. The interaction is governed by clear operational constraints—no final answers, no code writing or direct code explanation, and questions take precedence over assertions—and it proceeds incrementally through verifiable micro-steps, ideally with one question per turn. Metacognition is made explicit by comparing the initial and revised explanations and by forward planning in the *Feed-Forward* phase (and, in the extended version, *Transfer*). Finally, the agent leverages conversational memory to maintain coherence with the student's profile and history throughout the exchange. In its initial version, the protocol comprises five phases—*Feed-Up*, *Teach-Back*, *Probing*, *Refine*, and *Feed-Forward*—described below.

- **Feed-Up**

  In general, the Feed-Up phase sets the frame: the agent clarifies objectives, success criteria, and rules of engagement, making explicit that it will act as a dialogic tutor rather than an "answer engine." The goal is to align expectations and reduce ambiguity: what should the student be able to do or show at the end? Which parts of the reasoning will be observed? Which constraints will govern the dialogue?
  In the for-loop case studied, the aim is to explain how the construct works and to reason about the snippet that computes the sum from 1 to `n`. Minimum criteria are made explicit: distinguish the three elements of the `for` (initialization, condition, increment), state when the loop terminates, and identify the last value actually added. The agent may ask, for example: "By the end of this exchange, what will you be able to explain confidently about the `for` loop? Which event makes the loop stop?"

- **Teach-Back**

  In general, during Teach-Back the student speaks first and articulates their current understanding in their own words, without receiving immediate corrections. The agent listens and, if needed, asks micro-clarifications to complete the picture, but avoids introducing new content: the objective is to make the initial mental model visible as a basis for subsequent work.

  In the for-loop case, the agent invites the student to recount "what happens" from start to finish, including the role of `somma` and the meaning of the operation `somma += i`.

- **Probing**

  In general, Probing consists of targeted diagnostic questions that test specific aspects of the explanation, one at a time. The questions aim to surface gaps or common misconceptions, keeping cognitive load low and building evidence that prepares the reformulation; the goal is not to suggest the answer but to check, step by step, the robustness of the reasoning.

  In the for-loop case, typical questions concern the termination boundary and the last addend: "What is the last value of `i` that is actually added to `somma`? How do you infer this from the condition `i <= n`?"; "If you take a small case, e.g., `n = 3`, at which moments does `somma` change and which values does `i` take?" Asked one at a time, these checks help localize misconceptions precisely.

- **Refine**

  In general, Refine asks the student to reformulate their explanation by incorporating the evidence that emerged during probing. The goal is to foster self-correction and clarity: the agent does not provide the "right" version, but elicits a more precise synthesis, often with constraints on concision and a quick internal check on an example.

  In the for-loop case, the agent invites the student to rewrite the explanation in a few sentences, explicitly stating when the loop stops and what the last value added is. A quick verification can be added: "With `n = 3`, does your new explanation correctly predict how many times you enter the loop and which values you add to `somma`?" In this way the reformulation becomes traceable and anchored to concrete evidence.

- **Feed-Forward**

  In general, Feed-Forward projects learning into the future: the corrections just built are turned into control strategies and reusable checklists. The focus is on *how to verify* next time, rather than on memorizing a specific answer; the aim is transfer to related tasks.

  In the for-loop case, the agent asks the student to list two or three quick checks to apply the next time they encounter a loop, or proposes a minimal variation to test understanding ("If you changed <= to <, how would the last value added change?"), thus promoting transfer without leaving the problem's horizon.

## 4 Methods

Three simulated students were created with differing levels of prior understanding and cognitive style. The task for all students was to explain and reason about a basic C++ for loop that computes the sum from 1 to n. We conducted the simulations in ChatGPT using OpenAI's GPT-4o model (sessions: May–Sep 2025), following OpenAI's official model naming conventions. The agent autonomously generated and managed three distinct student profiles; all personas were simulated within a single chat session. This design choice—made to preserve turn-by-turn continuity while iterating on the protocol—introduces a risk of cross-persona carryover and demand characteristics. To bound that risk, each persona was initiated with an explicit role header and a full protocol reset (goals, constraints, tone), and the tutor restated the phase and rules at each switch. For each profile, the agent executed the full Teach-Back protocol in sequence, drawing on conversational memory to maintain within-interaction coherence. Human supervision was limited to providing the initial task and protocol instructions; all subsequent dialogue turns were produced directly by the model.

After iterative simulation, pedagogical review conducted by AI and human led to a revised version with added phases for "Slow Thinking" and "Transfer," and modified phase for "Incremental Probing." As a result the protocol is articulated into seven phases —*Feed-Up*, *Slow Thinking*, *Teach-Back*, *Probing-Incremental*, *Refine*, *Transfer Example*, and *Feed-Forward*— each with a distinct dialogic function described in Figure 1. In particular:

- **Slow Thinking**

  In general, the Slow Thinking phase introduces an explicit reflective pause before proceeding with questions or corrections. The goal is to reduce cognitive load and allow the student to reread their explanation, identify strengths and uncertainties, and formulate minimal self-checks. The agent invites the student to pause, mentally reformulate the key steps, and indicate where greater precision is needed, without adding new information.

  In the for-loop case, the pause guides the student to ask, for example, when the condition is tested, what the last value actually added is, and how the increment ultimately makes the condition false. The agent may prompt: "Without changing anything, reconstruct the case `n = 3` in your head: at which moments do `i` and `somma` change?"

- **Incremental Probing (revised)**

  In general, Probing is made incremental: one question at a time, each targeting a single micro-inferential step, so as to contain cognitive load and make the chain of reasoning traceable. The agent verifies, step by step, what has been established, avoids logical leaps, and builds local evidence before proceeding.

  In the for-loop case, the sequence may start from an operational detail ("What is the value of `i` at the first entry into the body?"), move to the timing of the check ("When is the condition evaluated relative to `somma += i` and `i++`?"), and reach the boundary ("After the last execution of `somma += i`, what is the value of `i`, and why does the condition become false at the next step?"). Targeted variants include micro-cases (`n = 1`, `n = 0`) or minimal modifications of the condition ("<" instead of "<="). Each answer is anchored to evidence ("show me on which line/moment it happens") before moving to the next question.

- **Transfer**

  In general, the Transfer phase aims to extend and generalize what has been learned beyond the specific problem, translating corrections into reusable strategies and testing them on minimal variations. The emphasis is on *how to check* in related situations rather than on recalling a single answer.

  In the for-loop case, the agent proposes small variations to verify the robustness of understanding: replace "<=" with "<", start from `i = 0` instead of `1`, use a decrement (`i-`), or sum only even numbers. The student is invited to predict the last value added and explain why, then to synthesize a personal checklist (e.g., alignment between condition and last addend; the order "test → body → increment"; verifying that the increment eventually makes the condition false). In this way, knowledge transfers to families of nearby tasks without leaving the problem's horizon.

This design makes the dialogic structure explicit, emphasizing cycles of articulation, challenge, and re-expression. The protocol thereby fosters active participation, encourages self-correction through questioning, and promotes metacognitive awareness by requiring learners to compare their initial and revised explanations. The refinement process, informed by the simulations, highlighted the value of an explicit Slow Thinking phase and incremental probing to reduce cognitive load, while the Transfer phase was introduced to extend learning beyond the immediate task.

A qualitative, interpretive analysis was conducted across the three simulations to identify recurring reasoning patterns, conceptual breakdowns, and moments of metacognitive insight. The process was grounded in close reading of the dialogues and informed by existing pedagogical models of feedback and regulation [14, 15]. Rather than applying a formal coding scheme, the analysis focused on extracting practical insights to guide protocol refinement—particularly regarding timing, cognitive load, and the effectiveness of dialogic scaffolding. In addition to close reading, we articulate a minimal analytic lens that informed our interpretation of the dialogues: *Clarification prompts* (requests for precision or missing details); *Misconception surfacing* (making a flawed idea explicit/salient); *Self-correction events* (student retracts/revises a prior claim); *Protocol-adherence markers* (moves consistent with the active phase); *Metacognitive statements* (reflection on process, monitoring, next steps). We did not conduct a formal coding procedure or report counts; this qualitative analytic lens is provided to make our qualitative judgments transparent and reproducible in future replications. To reduce construct confounds, we matched prompt length, role tone, and instruction density across conditions. Residual confounds (e.g., implicit metacognitive framing) are acknowledged in Limitations; future work includes isomorphic controls.

For comparison, we subsequently conducted additional baseline interactions with GPT-3.5 (free, not logged) in separate, newly initiated chats—both with and without the Teach-Back protocol—to

| Phase | Purpose (with Log Example) | Dialogic Function |
|---|---|---|
| Feed-Up | Clarify the learning goal and invite the student to explain. *E.g., Student A: "So, this code is used to sum all the numbers from 1 to n..."* | Starts from the learner's perspective, establishing ownership of the explanation. |
| Slow Thinking *(added)* | Encourage the student to pause before responding. *E.g., Student B gave a clearer, more structured explanation after being prompted to pause.* | Reduces impulsivity, supports deliberate formulation. |
| Teach-Back | Student explains concept as if teaching a peer. *E.g., Student A: "The for loop has three parts: initialization, condition, increment..."* | Externalizes current understanding, making reasoning visible. |
| Probing – Incremental *(revised)* | Ask diagnostic questions one at a time. *E.g., AI: "If n=0, does the loop run?" → Student C: "No, then it doesn't execute."* | Surfaces inconsistencies, promotes reasoning through guided questioning. |
| Refine | Student improves initial explanation. *E.g., Student B: "Okay, let me start over and try to be more precise..."* | Consolidates clarity, integrates corrections. |
| Transfer Example *(added)* | Apply the concept to a related task. *E.g., AI: "How would you change the loop if you wanted to sum only the even numbers up to n?"* | Tests generalization, promotes adaptive thinking. |
| Feed-Forward | Reflect on learning and plan next steps. *E.g., Student C: "I was wrong to say list... actually sum only holds the total."* | Makes metacognition explicit, reinforces self-regulation. |

Figure 1: Phases of the Teach-Back protocol as implemented by the AI tutor. The table summarizes each phase's pedagogical purpose—with brief log excerpts—and its dialogic function. Slow Thinking and Transfer were added during refinement, and Probing was revised to an incremental format (AI- and human-led pedagogical review).

prevent carryover effects. The GPT-3.5 contrast is illustrative and strictly within-model: it compares with-protocol vs. no-protocol dynamics while holding the same model (GPT-3.5) and setup fixed. We therefore do not draw cross-model conclusions (e.g., GPT-4o vs. GPT-3.5); interpretations are limited to qualitative differences attributable to the dialogic structure when the model is held constant. These results are presented in the *Illustrative Findings & Design Implications* section to highlight the added value of dialogic scaffolding.

*Appendix A* details the configuration prompt (main study), and *Appendix B* contains the logs of the three simulations (Students A, B, C) conducted with *GPT-4o*; *Appendix C* provides the configuration prompt, and *Appendix D* reports the logs of the three simulations in *GPT-3.5* (with- and without-protocol conditions).

## 5   Illustrative Findings & Design Implications

All three student simulations exhibited a progression from surface-level to more structured and accurate explanations. Diagnostic questioning helped reveal subtle misconceptions, while the "Refine" and "Feed-Forward" steps facilitated cognitive reorganization.

All students benefited; the specific patterns of improvement varied, underscoring the protocol's flexibility. Based on the simulation results, we integrated the following refinements into the protocol (derived post hoc and not exercised in the three simulations):

- Addition of a Slow-Thinking phase to prompt deliberate formulation
- Shift from batch to Incremental Probing
- Inclusion of a Transfer phase to prompt forward-looking application and self-explanation

These refinements aim to improve conceptual depth and foster reflective learning, and will be evaluated in the forthcoming Programming 1 classroom deployment.

### 5.1 Baseline Comparison: AI Without Protocol vs. AI With Teach-Back Protocol

To further validate the contribution of the protocol, we conducted a baseline comparison using a separate session with GPT-3.5. While the design and refinement of the protocol were carried out with GPT-4o, for the baseline, we used ChatGPT with GPT-3.5 (free, no login; Aug 2025); model availability in the free/no-login channel may vary over time and by region. This choice minimized contextual carryover between turns and provided a clean contrast between an AI acting without structured scaffolding and an AI guided by the Teach-Back method.

The purpose of this comparison was not to test raw model performance but to evaluate how the presence of a dialogic structure shapes the learning-oriented quality of the interaction. In the no-protocol baseline, the AI tended to give direct corrections and ready-made explanations. In contrast, the Teach-Back condition elicited a genuinely dialogic exchange in which the student was prompted to clarify, refine, and adapt their reasoning.

Across three simulated students, systematic contrasts emerged. Without protocol, students remained relatively passive, providing an initial explanation and then receiving corrective input. With the Teach-Back protocol, they became active participants, repeatedly reformulating and consolidating their understanding. Without protocol, error handling consisted of explicit corrections delivered by the AI; with protocol, the AI posed diagnostic questions that led students to self-correct. Metacognitive awareness was largely absent in the baseline, while the Teach-Back phases explicitly required students to reflect on and revise their explanations. Finally, the learning dynamic shifted from monologic correction to knowledge co-construction, with understanding emerging through questioning, reflection, and transfer. We organize the contrast along five dimensions: (1) interaction structure, (2) student role, (3) error handling, (4) metacognition, and (5) dialogic learning.

These differences are summarized in Figure 2, suggesting, in our qualitative contrast, that it is the protocol itself—rather than model scale—that enables deeper reasoning, dialogic engagement, and metacognitive learning.

## 6 Limitations

This is an exploratory study based on three AI simulations of the Teach-Back protocol on a single task (the C++ for loop) in AI self-play only. Strong assumptions: we treat simulated interactions as proxies for interactions with real students; we assume the selected task is representative and that the protocol sequence is transferable to other content areas. Robustness to assumption violations: results and interpretations may vary with different models/providers, sampling parameters, prompt wording/order, and—more broadly—with the "noise" typical of real classrooms (timing, distractions, incomplete inputs), which is absent from our simulations. Scope of claims: the evidence is descriptive and hypothesis-generating; we do not offer causal inferences nor generalizations beyond the domain and the three runs considered; we do not include a baseline (e.g., a human tutor) nor ablation studies of protocol components, and the refined seven-phase protocol was not separately evaluated. Factors influencing performance: model choice, sampling parameters (e.g., temperature, top-p), context length, prompt formulation, and the introduction of the Slow Thinking phase may materially affect outcomes; we ran simulations in a single chat session on GPT-4o, which can induce demand characteristics and cross-persona carryover. Efficiency and scalability: we do not analyze computational time/costs nor the scalability of the orchestration to large-scale scenarios (e.g., many sessions, token limits). Privacy and fairness: we do not use personal data; however, generative models may reflect biases that could be relevant in student-facing applications. We did not perform systematic

| Dimension | No Protocol (GPT-3.5 Free) | With Teach-Back Protocol (GPT-3.5 Free) |
|---|---|---|
| Interaction Structure | Flat Q&A: student explains once, AI corrects and extends. | Structured multi-phase dialogue (Feed-Up, Teach-Back, Probing, Refine, Feed-Forward, Transfer). |
| Student Role | Mostly passive: provides initial explanation, then listens to AI's corrections. | Active: repeatedly reformulates, clarifies, and applies understanding. |
| Error Handling | AI provides direct corrections and improved explanations. | AI asks diagnostic questions; student self-corrects through reflection. |
| Metacognition | Limited: occasional prompts, little explicit reflection. | Explicit: Feed-Forward requires student to reflect and restate understanding. |
| Dialogic Learning | Minimal dialogue, AI dominates with ready-made answers. | Strongly dialogic: iterative questioning and student-centered refinement. |

Figure 2: Within-model qualitative contrast (GPT-3.5). Comparison of no-protocol vs. with-protocol across five dimensions—interaction structure, student role, error handling, metacognition, dialogic learning. No cross-model inferences are claimed.

qualitative coding or intercoder reliability; claims remain descriptive and based on close reading guided by the qualitative analytic lens presented above. These limitations inform our future work: a human-tutor baseline, protocol ablations, additional tasks and disciplines, sensitivity analyses of key parameters, and deployment with real students under ethics approval. Additionally, we will formalize a lightweight analytic codebook with simple descriptive indicators (e.g., counts of clarifications, self-corrections, metacognitive statements) to strengthen qualitative rigor. Despite these boundaries, the study offers a transparent protocol design and a set of testable hypotheses about AI-mediated Teach-Back in introductory programming. To address the limitations above, our next phase includes validation with real students under ethics approval, a human-tutor and Socratic Q&A baseline, protocol ablations, broader task/discipline coverage, sensitivity analyses to model and prompt parameters, and profiling of time/cost and scalability, alongside a fairness/bias audit.

## 7    Conclusion and Broader Implications

This study set out to investigate whether and how a generative AI agent could implement a Teach-Back protocol to scaffold conceptual and metacognitive learning. Through three simulations with AI-generated novice student personas in introductory programming, the AI agent effectively conducted all phases of the protocol, enabling the students to articulate, challenge, and refine their reasoning.

The results show that the AI was able to elicit progressive improvements in conceptual clarity by prompting explanation, probing with diagnostic questions, and encouraging students to reflect on their learning. In particular, conceptual misunderstandings—such as incorrect reasoning about boundary conditions—were addressed through targeted, incremental questioning. The Feed-Forward phase supported metacognitive awareness, as students explicitly recognized the shifts in their understanding.

Importantly, this study serves as a preliminary testbed for the planned implementation of the Teach-Back protocol in an actual university-level course in Programming 1 for first-year computer engineering students. By validating and refining the protocol in simulated form, we lay the groundwork for ethically and pedagogically sound AI integration in classroom practice.

This proof-of-concept illustrates the potential of AI agents as structured, dialogic learning partners—not as shortcuts to answers, but as tools that can activate and deepen student thinking. The

simulations revealed recurring conceptual errors (e.g., boundary conditions in loops), which were effectively addressed through incremental questioning. Metacognitive reflection emerged during the Feed-Forward phase, particularly when students compared their revised reasoning to initial assumptions.

Based on these insights, the protocol was refined to include a deliberate Slow Thinking phase and to split diagnostic questions into incremental prompts. These adaptations helped reduce cognitive overload and promote conceptual clarity. In practice, we operationalize the protocol via one-question-at-a-time probing and a brief reflective cue, followed by small boundary-case checks and a student-written transfer checklist.

Crucially, the baseline comparison with GPT-3.5 (free, not logged) demonstrated that without structured scaffolding, students tended to remain passive, with the AI dominating the explanation. In contrast, the Teach-Back protocol fostered dialogic engagement, metacognitive reflection, and knowledge co-construction. This contrast reinforces the central claim of the study: it is not the model itself but the dialogic structure that transforms AI from an answer engine into a learning partner.

The results highlight the value of structured, agent-mediated dialogic learning protocols in early computer science education. Simulated interactions, even in the absence of real learners, exposed common reasoning pitfalls and informed iterative refinements to the protocol. This proof of concept supports the development of AI agents as dialogic partners—designed not to replace thinking but to cultivate it. The Teach-Back-based approach scaffolded both conceptual understanding and metacognitive awareness, emphasizing the importance of structured interaction design in avoiding over-assistance and promoting student-driven reasoning.

## AI Agent Setup

**Models.** Main simulations with OpenAI GPT-4o (May–Sep 2025); baseline with ChatGPT-3.5 (free, not logged).
**Interface/Orchestration.** Chat-based sessions in ChatGPT; GPT-4o personas were run within a single session (to preserve turn-by-turn continuity), while GPT-3.5 baselines used fresh, independent chats (with- and without-protocol).
**Tools/Retrieval.** None (no plug-ins, no external retrieval).
**Prompts.** Full operational prompts in Appendix A (main) and Appendix C (baseline); complete anonymized logs in Appendices B/D.
**Controls.** Dialogic constraints (no answer-dumping; questions over statements), phase sequencing, incremental probing; explicit role headers and per-phase restatements when switching personas.
**Parameters.** Default ChatGPT UI settings (temperature/top-$p$ not manually overridden); context limited to each chat.
**Analysis.** Close reading; no systematic coding; claims are descriptive/hypothesis-generating.
**Reproducibility.** Verbatim prompts and logs released; session design, model, interface, and dates reported for independent replication.

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

## Appendix A: Operational Prompt and Configuration (GPT-4o, Main Study)

**Prompt (verbatim)**

```
[START OF PROMPT]
You are an AI tutor helping a programming student consolidate their
understanding using the Teach-Back protocol.
Role: Dialogic AI tutor, not an answer engine.  Style: Supportive,
Socratic, focused on clarification, error detection, and reflection.
Constraints:  Do not provide final answers; do not directly write or
explain the student's code; prefer questions over statements.
Core tasks:  - Orchestrate the Teach-Back phases.  - Ask incremental,
targeted diagnostic questions (one per turn where possible).  -
Elicit metacognition and self-correction.  - Adapt tone and
questioning to the student's style.  - Follow the logic of the
Teach-Back protocol without enforcing rigid scripts.
Protocol phases to follow in each simulation:
  1. Feed-Up:  introduce the task with clear goals and friendly
     instructions.
  2. Teach-Back 1:  student provides a free, stepwise explanation.
  3. Probe & Misconceptions:  ask one incremental diagnostic at a
     time targeting likely errors.
  4. Refine:  ask the student to reformulate/clarify focusing on what
     probing revealed.
  5. Feed-Forward:  brief guided metacognitive reflection ("what
     changed?", "what will you do differently next time?").
[END OF PROMPT]
```

**Out-of-prompt notes**

- Simulations are multi-turn; each (simulated) student shows an internal evolution across turns.
- The agent was not configured via API or temperature settings; it operated via reasoned simulation in ChatGPT using **GPT-4o**.
- Simulations conclude with a brief qualitative analysis of strengths, limitations, and suggestions to improve the protocol.

## Appendix B: Simulated-student–AI Logs (GPT-4o, Main Study) (EN translated)

This appendix provides the English translations of the three simulated tutoring logs used in our study. The sessions were generated with an AI model (GPT-4o; provider: OpenAI) to enact the Teach-Back protocol on an introductory C++ for-loop task. The transcripts are offered to support transparency and enable independent inspection of interaction dynamics and prompt effects.

### Student A — Teach-Back Protocol (topic: for loop)

Student A Profile: First-year Computer Engineering; has seen *for* loops but struggles to explain them without examples; tends to give brief answers and improves with targeted prompts.

**[Feed-Up] [AI Tutor]**
Hi! In this exercise you will teach me how a *for* loop works in C++: I'm just a beginner AI!
Goal: explain what a *for* loop does and how the following code works:

```
int somma = 0;
for (int i = 1; i <= n; i++) {
    somma += i;
}
```

A good explanation: is clear and complete (use your own words); does not just read the code; includes what happens step by step; gives simple examples.

When you're ready, start your first explanation.

**[Teach-Back] [Simulated Student A]**

So, this code is used to add up all the numbers from 1 to n. The variable `somma` starts at zero, then there's the *for* loop that increases `i` from 1 to n. Each time it adds `i` to `somma`.

**[Probe & Misconceptions] [AI Tutor]**

Thanks for the explanation! Your description is broadly correct, but I'd like to understand a few points better: What happens if `n = 0`? Does the loop start anyway? How exactly does the *for* loop work? Can you describe the three parts (initialization; condition; increment)? You said that `somma` "adds `i`," but who actually does it? Where does the operation happen?

So far it seems that your explanation: covers the overall behavior, but doesn't explain how the *for* structure works, and doesn't consider edge cases. Would you like to reformulate and improve your explanation?

**[Refine] [Simulated Student A]**

Okay, I'll try again. The *for* loop has three parts: initialization: `int i = 1`, meaning `i` is created and set to 1; condition: `i <= n`, which is checked before each iteration; increment: `i++`, meaning `i` increases by one each time. If `n = 0`, the condition `1 <= 0` is false, so the loop does not start. During the loop, `somma += i;` means: take the current value of `somma` and add `i` to it, then store the result back in `somma`.

**[Metacognitive Feed-Forward] [AI Tutor]**

Great! You've significantly improved your explanation.

Now you have: described the full structure of the loop; clarified what happens step by step; considered an edge case.

One last question for you: How did this activity help you understand the *for* loop better? And what will you do differently the next time you use a loop?

**[Metacognitive Feed-Forward] [Simulated Student A]**

It helped me distinguish between the part of the code that starts the loop and the part that keeps it going. Before, I took everything for granted. Next time I'll try to think about how I would explain each part, even if I'm just writing for myself.

**Student B — Teach-Back Protocol (topic: for loop)**

Student B Profile: First year; has been studying C++ for 3 weeks; tends to be verbose and discursive but often mixes concepts; struggles to distinguish between "what the code does" and "how it does it"; more inclined to self-reflect and ask questions.

**[Feed-Up] [AI Tutor]**

Hi! Today you'll be the one explaining how the *for* loop works in C++.

Goal: help me understand how the code below works and why it is written this way:

```
int somma = 0;
for (int i = 1; i <= n; i++) {
    somma += i;
}
```

Reminder: speak as if you were explaining it to a classmate who didn't understand; don't read the code—explain it; you may use simple examples (e.g., "if n = 3 then...").

When you're ready, start your first explanation.

**[Teach-Back] [Simulated Student B]**

Okay, I'll try. So, the code is used to compute the sum of the numbers from 1 to n. To do this, we use a variable `somma` that starts at zero and then gets updated gradually. The *for* loop is the way you use in C++ to repeat something multiple times. In this case, the operation of adding `i` to `somma` is repeated. The loop starts at `i = 1`, then checks that `i <= n` and, if true, it executes `somma += i`, meaning it adds `i` to the variable `somma`. Then `i` increases by 1, and it starts over. In the end, `somma` will contain the result. I'm not sure if I explained everything well, but that's what I understood.

**[Probe & Misconceptions] [AI Tutor]**

You did a good job! I got the overall idea, but I still have a few doubts—could you help me clarify them? Why is `i` initialized to 1 and not 0? What happens if we use `i <= n` versus `i < n`? Does anything change in the result? You said that `somma += i` "updates" the variable, but what happens in detail?

Try to be more precise and imagine I've never seen a *for* loop. Could you reformulate your explanation taking these questions into account?

**[Refine] [Simulated Student B]**

Okay, I'll start over and try to be more precise. The *for* loop has three components: `int i = 1`: we create a variable `i` that we use for counting, and we start at 1 because we want to sum from 1 up to `n`. `i <= n`: this is the condition that is checked every time. If it's true, the loop continues. If we used `i < n`, then `n` would not be included in the sum! `i++`: every time the loop finishes one iteration, `i` increases by 1. The line `somma += i;` is equivalent to `somma = somma + i;`. So if `i = 3` and `somma = 6`, after this line `somma` becomes 9. The order of operations matters: first the condition is checked, then the code block is executed, then the increment happens. At the end of the loop, we have added all the numbers from 1 to `n`, one by one.

**[Metacognitive Feed-Forward] [AI Tutor]**

Excellent! You clarified many important points.

Your explanation is now: more structured; attentive to details (e.g., difference between <= and <); useful even for someone who doesn't already know the *for* loop.

Final question: What did you discover about how you reason when you explain? Did you notice anything you were taking for granted at the start?

**[Metacognitive Feed-Forward] [Simulated Student B]**

Yes, while I was explaining I realized that sometimes I skip steps in my head. For example, I had never really thought about why `i` starts at 1 or what happens if I use < instead of <=. In general, explaining made me realize that I only half-know things, and that saying them out loud forces me to put them in order.

**Student C — Teach-Back Protocol (topic: for loop, with conceptual error)**

Student C Profile: Confident but with a shallow understanding; uses technical language ("updates," "iteration") but confuses concepts; tends not to recognize their own errors immediately.

**[Feed-Up] [AI Tutor]**

Hi! In this exercise, you'll be the one explaining how this C++ code works:

```
int somma = 0;
for (int i = 1; i <= n; i++) {
    somma += i;
}
```

Goal: teach me what the *for* loop does and how it works.
Tips: speak as if I were a classmate who is struggling; explain what the lines do *and* why they work that way; if it helps, imagine an example with `n = 3` or `n = 5`.
Start whenever you're ready.

**[Teach-Back] [Simulated Student C]**

Okay. So, `somma` is initialized to 0, and then the *for* loop is used to create a list of numbers from 1 to `n`. Each time the loop runs, the number `i` is saved into `somma` so we can see all the values at the end. The instruction `i++` makes `i` keep increasing, so we take all the numbers from 1 to `n`. When the loop ends, we can use `somma` to see which numbers were used.

**[Probe & Misconceptions] [AI Tutor]**

Thanks for the explanation! You used a lot of technical terms—nice job. But a few doubts came up; can you help me understand better? You said that `somma` "saves all the numbers." But `somma` is a single variable: how can it save all the numbers? After the loop finishes, what does `somma` contain, for example if `n = 3`? If instead we wanted to see all the numbers from 1 to 3, how should we do it? Could you reformulate your explanation taking these questions into account?

**[Refine] [Simulated Student C]**
Hmm... it's true that `somma` is just one variable. So maybe I was wrong to say it "saves all the numbers." Rather than saving them, it adds them one by one, so `somma` changes value each time. For example, if `n = 3`, the values of `somma` would be: first 0; then 1 (0 + 1); then 3 (1 + 2); then 6 (3 + 3). In the end, `somma` equals 6. If we wanted to print the numbers, we should use `cout << i` inside the loop, not `somma`. Yes, that was my mistake.

**[Metacognitive Feed-Forward] [AI Tutor]**
Excellent! You recognized the ambiguity and clarified it very precisely. Now it's clear that `somma` does not store all the numbers, but only the running total updated each time.
Final question: What did you learn about how errors "work" when explaining? Do you often realize you've made an error while you're trying to teach something?

**[Metacognitive Feed-Forward] [Simulated Student C]**
Yes, it happens. When I explain something, sometimes I say things that "sound good," but then, as I'm saying them, I realize they don't add up. In this case I understood that I was confusing adding the numbers with storing them. It helped that you asked me what happens step by step. I'll do that too when I study.

## Appendix C: Operational Prompt (ChatGPT-4o → ChatGPT-3.5, Teach-Back Protocol)

This appendix presents the operational prompt generated by ChatGPT-4o based on the Teach-Back protocol developed in this study. The prompt was explicitly targeted to ChatGPT-3.5 (free version) and specified its role: *"You are an AI tutor helping a programming student consolidate their understanding using the Teach-Back protocol."* The full prompt is reproduced verbatim to support transparency and reproducibility.

**Full prompt (verbatim).**

```
You are an AI tutor helping a programming student consolidate their
understanding using the Teach-Back protocol.
Follow this 5-phase structure when interacting:
  1. FEED-UP Acknowledge the student's goal and ask them to explain
     the concept in their own words.  Do not give the explanation
     yourself.
  2. TEACH-BACK Let the student explain.  Then listen carefully and
     assess whether the explanation is clear, complete, and correct.
  3. PROBING & REFINEMENT If needed, ask one or more specific
     questions to help the student notice errors, refine vague ideas,
     or clarify incomplete reasoning.  Use diagnostic questions.  Do
     NOT just give the answer.
  4. FEED-FORWARD Ask the student to revise or re-express their
     explanation, based on what they realized.
  5. TRANSFER (Optional) If appropriate, ask the student how they
     would adapt the concept to a slightly different example or
     situation.

Your goal is to encourage thinking, clarification, and
self-correction, not to provide ready-made answers.  Keep your style
dialogic and student-centered.
Now wait for the student's explanation and begin the protocol.
```

## Appendix D: Simulated-student–AI Logs (ChatGPT3.5, baseline)(EN translated)

This appendix reports short comparative logs collected with and without the Teach-Back protocol using the same three simulated students considered in the main study. For each student, we reuse their first prompt to the ChatGPT-4o Dialogic-tutor-AI and record two conditions: (i) a no-protocol

baseline (free-form interaction), and (ii) a protocol-mediated interaction (Teach-Back cues). All interactions in this appendix were generated with ChatGPT-3.5 (OpenAI).

Each student's section presents: (A) No-Protocol Baseline — the response elicited by the student's first prompt without any scaffolding; (B) Teach-Back Protocol — the response when the same initial prompt is handled under the Teach-Back framing.

**Student A**

**No-Protocol Baseline**

**[Student A]**
I am a student. To check whether I've understood for loops, I have to explain what a for loop does and how the following code works:

```
int somma = 0;
for (int i = 1; i <= n; i++) {
    somma += i;
}
```

Here's my explanation: So, this code is used to add up all the numbers from 1 to n. The variable `somma` starts at zero, then there is the for loop that increases `i` from 1 to n. Each time it adds `i` to `somma`.

**[AI]**
Your explanation is clear and correct in its essential content! You've understood well what the `for` loop does in this case. Here are a few suggestions to improve it slightly and make it more precise (as a teacher might require):

Improved explanation: This code is used to compute the sum of all integers from 1 up to `n`.

1. Initialization: The variable `somma` is initialized to 0. It will be used to accumulate the result.
2. `for` loop: The variable `i` starts at 1 and increases by 1 at each iteration (`i++`) until it reaches `n` (inclusive, because the condition is `i <= n`).
3. Loop body: At each pass of the loop, the current value of `i` is added to `somma`, using the `+=` operator.

At the end of the loop, the variable `somma` contains the total of the numbers from 1 to `n`.

Practical example:
If `n = 4`, the loop will do:

- `somma = 0 + 1` → 1
- `somma = 1 + 2` → 3
- `somma = 3 + 3` → 6
- `somma = 6 + 4` → 10

So at the end `somma` will be 10, i.e., $1 + 2 + 3 + 4$.

If you want, I can also help you write a version of the explanation in a more "assignment-style" or "oral-exam" language.

**With protocol**

**[Student A]**
Hi! My task is to explain what a `for` loop does and how the following code works:

```
int somma = 0;
for (int i = 1; i <= n; i++) { somma += i; }
```

So, this code is used to add up all the numbers from 1 to n. The variable `somma` starts at zero, then there's the `for` loop that increases i from 1 to n. Each time it adds i to `somma`.

**[AI]**
Perfect, you're at the right point in the protocol: TEACH-BACK.
You said that:

- The code is used to add up all the numbers from 1 to `n`.

- The variable `somma` starts at 0.

- The `for` loop makes i increase from 1 to `n`.

- At each iteration it adds i to `somma`.

You've captured the general idea and the basic functioning. Now let's move to the PROBING & REFINEMENT phase, to make sure everything is solid. Here are some questions to clarify and refine:

- Are you sure that i reaches exactly `n`? In what cases might it not?

- What happens if `n` equals 0? Is the loop executed?

- And if `n` is negative?

- Why does i start at 1 and not at 0? How would the result change if it started at 0?

Feel free to answer out loud or in writing. Let's shed light together on any uncertainties.

**[Student A]**
Okay, I'll try again. The `for` loop has three parts:

- initialization: `int i = 1`, that is, i is created and has value 1;

- condition: `i <= n`, which is checked before every iteration;

- increment: `i++`, meaning i increases by one each time.

If `n = 0`, the condition `1 <= 0` is false, so the loop doesn't start. During the loop, `somma += i;` means: take the current value of `somma` and add i to it, then store the result in `somma`.

**[AI]**
Great work! You're in the FEED-FORWARD phase, where you've already clarified several points very precisely. You have:

- broken down the `for` loop into its three fundamental components (initialization, condition, increment);

- correctly recognized that if `n = 0`, the initial condition is false and the loop does not start;

- correctly explained the meaning of `somma += i`.

You're showing a good level of understanding. I now propose a small step forward with a TRANSFER question, to check how well you can adapt this concept. How would you change the loop if you wanted to sum only the even numbers from 1 to `n`? Try to explain it in your own words and, if you want, also with a small piece of code.

**Student B**

**No-Protocol Baseline**

**[Student B]**
I am a student. To check whether I have understood for loops well, I have to explain what a `for` loop does and how the following code works:

```
int somma = 0;
for (int i = 1; i <= n; i++) { somma += i; }
```

Here is my explanation: So, the code is used to calculate the sum of the numbers from 1 to n. To do this, a variable `somma` is used, which starts at zero and then gets updated gradually. The `for` loop is the way used in C++ to repeat something multiple times. In this case, the operation of adding `i` to `somma` is repeated. The loop starts from `i = 1`, then it checks that `i <= n` and, if that's true, it executes `somma += i`, that is, it adds `i` to the variable `somma`. Then `i` increases by 1, and it starts over. In the end, `somma` will contain the result. I'm not sure if I explained everything well, but that's what I understood.

**[AI]**
You did an excellent job explaining the code! Your explanation is clear and shows that you've understood the basic concept of the `for` loop.

I'll just give you a few suggestions to make it even more precise and complete:

Your code:

```
int somma = 0;
for (int i = 1; i <= n; i++) {
    somma += i;
}
```

Improved explanation:
The code is used to calculate the sum of the integers from 1 to n, inclusive. You start by declaring a variable `somma`, initialized to 0. This variable will be used to accumulate the result.

Then you use a `for` loop, which is a control structure in C++ used to repeat a block of statements a certain number of times.

In this case:

- You start from `i = 1`;
- You continue the loop as long as `i <= n`;
- On each pass of the loop, you execute `somma += i`, i.e., you add the current value of `i` to the variable `somma`;
- Then `i` is automatically increased by 1 (`i++`).

When `i` becomes greater than `n`, the loop stops and the variable `somma` contains the desired result, i.e., the sum of the numbers from 1 to `n`.

Practical example:
If `n = 5`, the loop will perform:

- `somma = 0 + 1 = 1`
- `somma = 1 + 2 = 3`
- `somma = 3 + 3 = 6`
- `somma = 6 + 4 = 10`
- `somma = 10 + 5 = 15`

In the end, `somma` will be **15**, i.e., `1 + 2 + 3 + 4 + 5`.

Conclusion:
You've understood it well and you're on the right track. If you want to improve a bit more, you can also try to explain what happens step by step or talk about the three main components of the `for`:

```
for (initialization; condition; update)
```

In your case:

- Initialization: `int i = 1`
- Condition: `i <= n`

- Update: `i++`

If you can master this as well, you're already at a good level! Do you want to try rewriting it with these added details?

**With protocol**

**[Student B]**
Hi! My task is to explain what a `for` loop does and how the following code works:

```
int somma = 0;
for (int i = 1; i <= n; i++) { somma += i; }
```

So, the code is used to calculate the sum of the numbers from 1 to n. To do this, a variable `somma` is used that starts at zero and then gets updated gradually.

The `for` loop is the way used in C++ to repeat something multiple times. In this case, the operation of adding `i` to `somma` is repeated.

The loop starts from `i = 1`, then it checks that `i <= n` and, if true, it executes `somma += i`, that is, it adds `i` to the variable `somma`.

Then `i` increases by 1, and it starts over.

In the end, `somma` will contain the result.

I'm not sure if I explained everything well, but that's what I understood.

**[AI]**
TEACH-BACK Thanks! You did a good job describing what the code does and how the `for` loop works in general terms. It's clear that you understood that:

- the loop is used to iterate from 1 to `n`;

- each time it updates the variable `somma` by adding `i`;

- and that at the end of the loop, `somma` contains the sum of the numbers from 1 to `n`.

Now let's take a moment for PROBING & REFINEMENT. You said: "The loop starts from `i = 1`, then it checks that `i <= n` and, if true, it executes `somma += i`... Then `i` increases by 1, and it starts over."

Diagnostic question: At exactly what moment is `i` increased by 1? Before or after executing `somma += i`?
And another to go deeper: Can it happen that the body of the loop (`somma += i`) is never executed? Under what conditions would that occur?
Answer these two questions to clarify your understanding even further."

**[Student B]**
Okay, I'll start over and try to be more precise. The `for` loop has three components:

- `int i = 1`: a variable `i` is created to count, and we start from 1 because we want to sum from 1 up to `n`;

- `i <= n`: this is the condition that is checked every time. If it's true, the loop continues. If we used `i < n`, then `n` would not be included in the sum!;

- `i++`: every time the loop finishes one round, `i` increases by 1.

The line `somma += i;` is equivalent to `somma = somma + i;` so if `i = 3` and `somma = 6`, after this line `somma` becomes 9.

The order of things is important: first the condition is checked, then the code block is executed, then the increment happens.

At the end of the loop, we have added all the numbers from 1 to `n`, one by one.

**[AI]**
FEED-FORWARD Great work! Your explanation is now much more precise and shows a solid understanding of both the structure of the `for` loop and the order of execution: condition → loop body → increment. You also clearly explained the importance of the comparison `i <= n` (including n) and the meaning of the operation `somma += i`.

To consolidate even more: TRANSFER If instead you wanted to sum only the even numbers from 1 to n, how would you modify the `for` loop? You can propose a possible solution and explain why it would work.

**Student C**

**No-Protocol Baseline**

**[Student C]**
I am a student. To check whether I've understood for loops, I have to explain what a `for` loop does and how the following code works:

```
int somma = 0;
for (int i = 1; i <= n; i++) { somma += i; }
```

Here's my explanation: So, `somma` is initialized to 0, and then the `for` loop is used to create a list of numbers from 1 to n.

Each time the loop runs, the number `i` is saved inside `somma` so that we can see all the values at the end. The instruction `i++` makes `i` keep increasing, so we take all the numbers from 1 to n.

When the loop finishes, we can use `somma` to see which numbers were used.

**[AI]**
You made a good attempt, and it's clear you understood the general idea of the `for` loop, but there are a few small errors or imprecisions in the explanation. I'll help you correct it step by step to make it clearer and more precise. Code to explain:

```
int somma = 0;
for (int i = 1; i <= n; i++) {
    somma += i;
}
```

What this code does:

- `int somma = 0;` Initializes a variable called `somma` to 0. It will be used to hold the result of the total sum.
- `for (int i = 1; i <= n; i++)` This is a loop that starts from `i = 1` and continues as long as `i` is less than or equal to `n`, increasing `i` by 1 each time.
- `somma += i;` At each iteration of the loop, it adds the value of `i` to the variable `somma`.

What the code actually does:
The program computes the sum of the integers from 1 to n. For example, if `n = 5`, the code computes: `somma = 1 + 2 + 3 + 4 + 5 = 15`.

Corrections to your explanation:

- You say the loop "creates a list of numbers from 1 to n" → in reality it doesn't create any list; it is just using the values from 1 to `n` one at a time to add them up.
- You say "the number `i` is saved inside `somma`" → `i` isn't being saved; rather, the value of `i` is added to `somma`.
- You say "so we can see all the values at the end" → at the end we only see the total sum, not the individual values.

A correct and simple explanation:

> The variable `somma` starts at 0. The `for` loop sets the variable `i` to 1 and increases it by 1 each time, up to `n`. Each time the loop runs, the value of `i` is added to `somma`. At the end of the loop, `somma` contains the sum of all the numbers from 1 to `n`.

If you want, I can also help you write it in a "to-hand-in" style for school, or ask you some questions to check whether you've really understood. Would you like that?"

**With protocol**

**[Student C]**
Hi! My task is to explain what a `for` loop does and how the following code works:

```
int somma = 0;
for (int i = 1; i <= n; i++) { somma += i; }
```

So, `somma` is initialized to 0, and then the `for` loop is used to create a list of numbers from 1 to n. Each time the loop runs, the number `i` is saved inside `somma` so that we can see all the values at the end. The instruction `i++` makes `i` keep increasing, so we take all the numbers from 1 to n.

When the loop ends, we can use `somma` to see which numbers were used.

**[AI]**
TEACH-BACK Thanks! You explained the code confidently and the general idea of the `for` loop. You correctly identified the role of `somma`, the initialization, the condition `i <= n`, and the increment `i++`. That said, there are a few points worth clarifying: You say the loop 'is used to create a list of numbers from 1 to n,' but there is no list in the code. You said that `i` 'is saved inside `somma` so we can see all the values at the end.' It is true that `i` is added, but `somma` does not store each value, only the total result. Finally, you said that 'we can use `somma` to see which numbers were used.' Perhaps you meant something else?

**PROBING & REFINEMENT** I'll ask a few questions to help you refine your explanation: When you say the loop 'creates a list,' do you mean the values are stored somewhere, or only visited one at a time? What exactly does the variable `somma` contain at the end of the loop? Is it a list of numbers, or something else? If `n` were 5, what would the values of `i` be during the loop, and what would `somma` be at the end? Take a moment to reflect and then answer.

**[Student C]**
Hmm… it's true it's not correct to say a list; `somma` is just a variable. So maybe I was wrong to say that it 'saves all the numbers.'

Rather than saving them, it adds them one by one, so `somma` changes value each time.

**[AI]**
FEED-FORWARD Excellent rethinking! You've captured the point perfectly: the variable `somma` doesn't store all the numbers; it keeps the running total as the loop progresses. Now I'm asking you to reformulate your entire explanation, as if you wanted to clarify it to someone who misunderstood it the first time. Focus on the idea of the loop, the role of `i`, and the meaning of `somma`. Go!

## Appendix E: Responsible AI Statement

This study involves only AI-simulated tutoring sessions; no human participants were involved, and no personal data were used. The work is conceived as a transparent, design-led proof of concept that illustrates how a dialogic, protocol-driven agent can scaffold conceptual clarification and metacognitive reflection in introductory programming.

To mitigate risks of over-assistance and cognitive offloading, the agent operated under explicit constraints and a structured pedagogy: it did not provide final answers, did not write or directly explain the student's code, and prioritized questions over statements. The refined protocol introduced a deliberate Slow Thinking phase, an Incremental Probing and a Transfer phase to manage cognitive load and promote self-correction through targeted, one-step-at-a-time diagnostic questions. Prompts and anonymized logs are released in the appendices to enable inspection and reuse, supporting transparency and responsible evaluation.

We acknowledge that generative models may reflect biases that could be relevant in student-facing applications. Future work explicitly includes classroom deployment under ethics approval, alongside a fairness/bias audit, human-tutor and Socratic baselines, and ablations of protocol components to probe robustness and limitations.

## Appendix F: Reproducibility Statement

We provide in (Appendix A-B-C-D) verbatim operational prompts and complete anonymized transcripts of the simulated sessions to facilitate inspection and replication. The main study used ChatGPT with **GPT-4o** to enact the Teach-Back protocol (sessions: May–Sep 2025); a baseline comparison was later conducted with **ChatGPT 3.5 (free, no login)**, with and without the protocol cues, reusing the same initial prompts. We follow OpenAI's official model naming conventions.

No model training, fine-tuning, or parameter tuning was performed; the agent was not configured via API or temperature settings and operated within ChatGPT. The task, protocol phases, and evaluation focus are described in the main text. To reproduce the qualitative findings, run the verbatim prompts on the stated models, carry out the full multi-turn protocol on the same for-loop task, and compare the resulting dialogues along the dimensions discussed. (interaction structure, student role, error handling, metacognition, dialogic learning). Code release is not applicable to this study; all materials needed for qualitative replication (prompts and logs) are included in the appendices.


